# Meropenem Stability in Human Plasma at −20 °C: Detailed Assessment of Degradation

**DOI:** 10.3390/antibiotics10040449

**Published:** 2021-04-16

**Authors:** Matthias Gijsen, Benjamin Filtjens, Pieter Annaert, Yeghig Armoudjian, Yves Debaveye, Joost Wauters, Peter Slaets, Isabel Spriet

**Affiliations:** 1Clinical Pharmacology and Pharmacotherapy, Department of Pharmaceutical and Pharmacological Sciences, KU Leuven, 3000 Leuven, Belgium; isabel.spriet@uzleuven.be; 2Pharmacy Department, UZ Leuven, 3000 Leuven, Belgium; 3Department of Mechanical Engineering, KU Leuven, 3000 Leuven, Belgium; benjamin.filtjens@kuleuven.be; 4Department of Electrical Engineering (ESAT), KU Leuven, 3000 Leuven, Belgium; peter.slaets@kuleuven.be; 5Drug Delivery and Disposition, Department of Pharmaceutical and Pharmacological Sciences, KU Leuven, 3000 Leuven, Belgium; pieter.annaert@kuleuven.be; 6BioNotus, Galileilaan 15, 2845 Niel, Belgium; yeghig.armoudjian@bionotus.com; 7Laboratory for Intensive Care Medicine, Department of Cellular and Molecular Medicine, KU Leuven, 3000 Leuven, Belgium; yves.debaveye@uzleuven.be; 8Laboratory for Clinical Infectious and Inflammatory Diseases, Department of Microbiology, Immunology and Transplantation, KU Leuven, 3000 Leuven, Belgium; joost.wauters@uzleuven.be

**Keywords:** meropenem, stability, degradation, plasma

## Abstract

There are concerns about the stability of meropenem in plasma samples, even when frozen at −20 °C. Previous smaller studies suggested significant degradation of meropenem at −20 °C after 3–20 days. However, in several recent clinical studies, meropenem plasma samples were still stored at −20 °C, or the storage temperature and/or time were not mentioned in the paper. The aim of this study was to describe and model meropenem degradation in human plasma at −20 °C over 1 year. Stability of meropenem in human plasma at −20 °C was investigated at seven concentrations (0.44, 4.38, 17.5, 35.1, 52.6, 70.1, and 87.6 mg/L) representative for the range of relevant concentrations encountered in clinical practice. For each concentration, samples were stored for 0, 7, 14, 21, 28, 42, 56, 70, 84, 112, 140, 168, 196, 224, 252, 280, 308, 336, and 364 days at −20 °C before being transferred to −80 °C until analysis. Degradation was modeled using polynomial regression analysis and artificial neural network (ANN). Meropenem showed significant degradation over time in human plasma when stored at −20 °C. Degradation was present over the whole concentration range and increased with higher concentrations until a concentration of 35.1 mg/L. Both models showed accurate prediction of meropenem degradation. In conclusion, this study provides detailed insights into the concentration-dependent degradation of meropenem in human plasma stored at −20 °C over 1 year. Meropenem in human plasma is shown to be stable at least up to approximately 80 days when stored at −20 °C. The polynomial model allows calculating original meropenem concentrations in samples stored for a known period of time at −20 °C.

## 1. Introduction

Meropenem is a β-lactam antibiotic with broad-spectrum activity against both Gram-negative and Gram-positive bacteria. Consequently, meropenem is frequently used to treat infections in critically ill patients. Meropenem pharmacokinetics has been shown to be highly variable in critically ill patients, often resulting in poor pharmacokinetic/pharmacodynamic (PK/PD) target attainment in critically ill patients. Therefore, recent guidelines recommend therapeutic drug monitoring for meropenem (and other β-lactams) to improve PK/PD target attainment in critically ill patients [1,2].

Over the last decade, many studies have evaluated meropenem pharmacokinetics, using plasma samples collected from patients treated with meropenem. In many hospitals, quantification of meropenem is still not part of routine laboratory testing [3]. As a result, meropenem samples collected during clinical studies are often processed and frozen until shipment and/or batch analysis. Studies have reported that β-lactams show substantial degradation in plasma depending on the storage temperature and storage time [4,5,6]. For meropenem, significant degradation has been shown beyond 6 h at 24 °C or 3 days at 4 °C [7]. Hence, plasma samples should be processed and frozen as soon as possible.

Even when plasma samples are frozen, there is uncertainty about meropenem long-term stability. Stability at −20 °C has only been demonstrated for 3 days up to 20 days [7,8,9,10]. Mortensen et al. [7] recommend that plasma samples for quantification of meropenem stored for longer than 3 days should be kept at −80 °C. To date, stability of meropenem in human plasma at −80 °C has been shown for at least 3–9 months [5,6,11]. However, in several PK studies, meropenem plasma samples were stored at −20 °C for longer than 3 days, or the storage time was not mentioned in the paper [12,13,14,15]. Furthermore, multiple studies did not explicitly mention storage temperature or storage time [16,17,18,19].

The aim of this study was to describe and model meropenem degradation in human plasma at −20 °C over 1 year.

## 2. Results

In total, 456 plasma samples were prepared and stored at −20 °C for various durations of time. None of the 57 blank plasma samples showed a detectable meropenem concentration (lower limit of quantification = 0.1 mg/L). For the 399 plasma samples spiked with meropenem, mean meropenem plasma concentrations (*n* = 3) for seven different spiked concentrations stored during 19 different storage times at −20 °C are shown in Table 1. The measured concentration for plasma samples that were stored during the complete study period at −80 °C (i.e., time at −20 °C = 0 days) was within 94–105% of the spiked concentration, thus showing no measurable degradation under these conditions. As demonstrated in Figure 1, meropenem showed significant degradation over time in human plasma when stored at −20 °C. Degradation was present over the whole concentration range and increased more than proportionally with higher spiked concentrations until a spiked meropenem concentration of 35.1 mg/L.

The polynomial model showed an accurate prediction of initial meropenem concentrations within the studied range, with a mean root-mean-square error (RMSE) of 4.10% and mean *R*^2^ of 0.932. The worst approximation was found for the lowest concentration, with an RMSE of 5.68% and *R*^2^ of 0.868. The evaluation metrics are summarized in Table 2. The regression equation for degradation at −20 °C was found to be
$$Y=0.779-0.166X_{1}-0.046X_{2}+0.004X_{1}^{2}-0.021X_{2}^{2}+0.019X_{1}X_{2},$$
where Y is the relative meropenem concentration remaining after degradation at −20 °C with respect to the spiked concentration at t0, $X_{1}$ is the standardized storage time at −20 °C (days), and $X_{2}$ is the standardized measured concentration (mg/L). The F-statistic was found to be 372.4 with a *p*-value of 4.8 × 10^−74^.

All coefficients were found to be significant, except for the second-order effect of storage duration. The regression coefficients are summarized in Table 3.

Figure 2 shows the percentage meropenem concentrations remaining as a function of storage time at −20 °C as simulated using the polynomial model, thereby confirming concentration-dependent degradation, with faster degradation at higher concentrations. At lower concentrations (0.44 and 4.38 mg/L), degradation became significant at 135 and 127 days (Appendix A, Appendix A), respectively. In contrast, with increasing concentrations, significant degradation occurred earlier. This effect appeared to stabilize at higher concentrations, resulting in the degradation becoming significant at around 80 days for concentrations above 35.1 mg/L (Appendix A, Appendix A). Noteworthy, at the highest concentration (i.e., for E7), meropenem showed lower relative degradation than for E5 and E6 after 100–150 days.

Accurate prediction of initial meropenem concentrations within the studied range was also obtained with the artificial neural network (ANN) model, with a mean RMSE of 3.55% and mean *R*^2^ of 0.945. The results suggest that the ANN outperformed the polynomial model over the studied concentration range. The evaluation metrics are summarized in Table 2. Figure 3 illustrates that the neural network was able to better capture the nonlinearity of the degradation profile, whereas the polynomial model was restricted to a quadratic approximation.

High correlation was observed between both regression models and the ground-truth experimental measurements, as shown in Figure 4; excellent correlation was observed for both models with Pearson correlation coefficients (95% CI) of 0.972 (0.960–0.980) and 0.969 (0.956–0.978) for the ANN and polynomial model, respectively. The *t*-tests were statistically significant for both models, with both models having a *p*-value of 2.2 × 10^−16^. The results, thus, suggest that the null hypothesis should be rejected (*p* < 0.05), i.e., the relationship between the models’ predictions and experimental measurements was statistically significant. The Bland–Altman plots indicate narrow limits of agreement (<10%), within the 15% range of accepted variability, for both models. The bias (95% CI) was found to be −0.092 (−0.775–0.592) and −0.032 (−0.751–0.686) for the ANN and polynomial model, respectively. No statistical evidence was found to reject the null hypothesis (*p* > 0.05), with a *p*-value of 0.791 for the ANN and 0.929 for the polynomial model, i.e., there was no statistically significant bias between the models’ predictions and the measurements.

### Data Availability

The dataset is available in Table 1. The polynomial meropenem degradation model, corresponding to the abovementioned equation, can easily be applied by using the documents made publicly available at https://github.com/BenjaminFiltjens/meropenem_degradation (accessed on 29 March 2021).

## 3. Discussion

In this degradation study, we present the detailed degradation profile of meropenem in human plasma over 1 year at −20 °C, within a range of clinically relevant concentrations. We showed that there is clear and significant degradation over time. Both models developed in this study describe meropenem degradation in human plasma over 1 year with good performance (RMSE < 5% and *R*^2^ > 0.9). As the polynomial model has been made publicly available, this can be used to investigate the relative meropenem concentration remaining in plasma samples stored at −20 °C. Hence, this allows back-calculation to initial meropenem concentrations.

On the one hand, this study confirmed degradation of meropenem at –20 °C in human plasma samples. On the other hand, detailed assessment over 1 year showed that degradation was not yet significant after 3–20 days, as suggested by previous studies [7,8,9,10]. Noteworthy, our results stand in contrast to the findings published by Zander et al. [11], who reported substantial decreases of meropenem concentration ≥7 days of storage at −20 °C, with minor amounts remaining after 90–180 days, in serum-based test samples. The authors used a novel evaluation protocol, in which substantial changes were defined as changes ≥15% and ≥3 coefficients of variation (CV) compared with baseline (and *p* < 0.01). In contrast, we used regression analysis as recommended by the ICH guideline for the evaluation of stability data [20]. We collected detailed degradation measurements (i.e., seven concentrations over 19 time points) over 1 year, thereby allowing reliable estimation of meropenem degradation. Interestingly, the other studies suggesting degradation at −20 °C after 3–20 days did not study degradation beyond this storage time [10] or, when they did, higher percentages of meropenem remaining were found than reported by Zander et al. [11] after 7–30 days [7,9]. The methodology used to describe degradation might be a reason for this difference in stability. The biological matrix studied might also be an important factor (plasma vs. serum). Moreover, all of these studies investigated only up to three concentrations in duplicate or triplicate and over a smaller range of concentrations. For the lower concentrations investigated in the present study (i.e., 0.44 and 4.38 mg/L), representative for trough concentrations encountered in clinical practice, degradation was only significant after 127–135 days storage at −20 °C. Interestingly, for the lowest concentration (i.e., 0.44 mg/L), which was close to lowest level of quantification, changes in concentration up to ±20% might be considered acceptable, as recommended in guidelines for bioanalytical method validation [21]. As a result, for very low meropenem concentrations, degradation might only become significant after 176 days (i.e., critical threshold value for 80% degradation of 0.44 mg/L). Importantly, we showed concentration-dependent degradation in plasma; hence, for higher meropenem concentrations (i.e., 35.1–87.6 mg/L), degradation became significant when stored for approximately 80 days at −20 °C. As shown in Figure 2, the relative degradation for the highest concentration (i.e., 87.6 mg/L) decreased after 100–150 days and became lower than for the next two concentrations (i.e., 70.1 and 52.6 mg/L). This decrease seemed to compensate for higher relative degradation during the first 100 days. Considering all of the above, meropenem seems to be stable in plasma at −20 °C for longer than the suggested 3–20 days [7,8,9,10].

Two types of modeling strategies were used to describe meropenem degradation in plasma at −20 °C, both leading to accurate prediction of initial meropenem concentrations. Compared with the polynomial model, the ANN showed a better approximation of the degradation profile over the studied concentration range. The better predictive performance of the ANN can be attributed to its universal ability to approximate any nonlinear function, whereas the polynomial model was restricted to a quadratic approximation. However, the predicted degradation of the polynomial model also showed excellent correlation with the experimental measurements, with predictions falling within the range of expected analytical variability. The difference in predictive ability of both models is, in our opinion, not clinically relevant; as shown in the Bland–Altman plot (Figure 4), limits of agreement lie within 10% for both models. The polynomial model has the advantage that it is able to outline the contribution and interaction between the predictors and the dependent variable. Accordingly, along with the fact that the polynomial equation can easily be made publicly available, we are in favor of using the polynomial model for approximation of meropenem degradation in plasma samples stored at −20 °C. 

The approximation of the relative meropenem concentration remaining, according to the polynomial model, showed the largest error at the lowest concentration (i.e., 0.44 mg/L). However, this error was still relatively low and within the range of expected analytical variance (i.e., 15% of the measured concentration). Moreover, for the next concentration (i.e., 4.38 mg/L), the error was already much lower, which is reassuring for clinical decisions made on the basis of concentrations measured in this range of concentrations. The leave-one-experiment-out cross-validation warrants generalization of the polynomial model to all meropenem concentrations within the experimentally validated range. However, caution is warranted if the pretrained model is used on measurements outside the experimentally validated range.

Until now, it remains unclear which mechanism is responsible for this ongoing degradation at −20 °C. Chemical hydrolysis is known to be an important determinant for meropenem stability in aqueous solutions at 25 °C and under simulated physiologic conditions at 37 °C, such as in plasma [22,23]. Higher temperature (45 °C) has been shown to increase the degradation rate [22]. Consequently, this chemical hydrolysis decreased at lower temperatures. Nevertheless, it appeared still active at −20 °C as demonstrated by significant degradation over time.

This study illustrates the importance of reporting detailed storage conditions. Storage of samples at −20 °C might lead to an underestimation of the initial concentration, thus leading to possibly erroneous conclusions such as pharmacokinetic/pharmacodynamic target non-attainment and dose increase. For studies during which samples have been stored at −20 °C, researchers can use the easily accessible degradation model (see Section 2) to calculate initial meropenem concentrations. This might lead to different conclusions, such as higher target attainment or lower clearance values. As an illustration, in a recent pharmacokinetic study [13], plasma samples were stored at −20 °C (duration unknown). Taking into account a target of 100% *f*T > MIC (free time above the minimal inhibitory concentration), and using the EUCAST MIC clinical breakpoint for susceptibility to Enterobacterales of 2 mg/L, the conclusion for target attainment would be different if samples were stored at −20 °C for 3 months or longer (for the maximum trough concentration). Another recent study reported higher meropenem clearance than in healthy volunteers, which was not found to be related with creatinine clearance (which is known to be the major driver for meropenem clearance) [12]. A possible explanation for this might be degradation of meropenem in the plasma samples which were stored at −20 °C (for an unknown period of time). Consequently, similar to external quality control programs that are in place for antimicrobial therapeutic drug monitoring [24], reporting storage conditions is equally important and relevant for studies measuring meropenem in plasma. This is an important issue that needs to be addressed by reviewers. For meropenem, many papers failed to mention storage temperature [12,13] and, even when mentioned, the storage time was often not reported [16,17,18,19]. 

For the studies where meropenem was stored at −20 °C, the model provided in this study allowed an evaluation of the impact of degradation at −20 °C on the conclusions of the study. According to the relative meropenem concentration remaining after degradation, it is recommended to reassess the results and investigate if the conclusions remain unchanged.

This study has several strengths. First, this study presented a detailed description of meropenem degradation at −20 °C, based on a large number of samples. These samples covered an extended period of time, i.e., 1 year, which was well represented by 19 different time points. Additionally, these samples covered a broad range of clinically relevant concentrations. Second, we developed a model that approximates the relative meropenem concentration over time with very good accuracy. Third, we made the polynomial model publicly available, making it an easily applicable tool for other researchers. With this model, estimation of the initial meropenem concentration is possible if (1) the degraded concentration can be measured, and (2) the storage duration at −20 °C is known. Fourth, we illustrated the applicability of ANN modeling for this type of research. Lastly, we confirmed stability of meropenem in human plasma up to 1 year at −80 °C.

This study also has several limitations that need to be acknowledged. First, the model developed in this study was not externally validated. Hence, caution should be taken when applying the model before it has been externally validated. Ideally, native plasma samples from critically ill patients receiving meropenem are used for this validation. Second, in the current study, the metabolic and hydrolytic activity of the plasma used was not verified, mainly because our stability assay was performed at −20 °C (instead of 37 °C for the ‘classical’ plasma stability assay). Residual enzymatic or chemical hydrolytic capacity of the plasma at −20 °C could have been evaluated by using compounds with known (in)stability at −20 °C as positive and negative controls [25]. However, enzymatic hydrolysis is unlikely for meropenem in the absence of beta-lactamase enzymes. Hence, we expected mainly chemical hydrolysis at a rate which was not expected to vary substantially between different batches of plasma, especially when studied at −20 °C. Additionally, there was likely only minor influence from variations in pH and co-administered drugs, as plasma used in the present study was collected from healthy volunteers. Third, the degradation of meropenem over time showed variation at different time points, as can be seen in Figure 1. There was no apparent reason for this variation, as sample preparation and bioanalysis were thoroughly reviewed by the research team. Despite this variation, the degradation shows a clear and consistent trend, as demonstrated by the good accuracy and fit of the model. 

In conclusion, this study provided detailed insights into the concentration-dependent degradation of meropenem in human plasma stored at −20 °C over 1 year. Meropenem in human plasma was shown to be stable at least up to approximately 80 days when stored at −20 °C. Hence, during study design, researchers should take this stability information into account when planning storage and analysis of meropenem plasma samples. The degradation model developed in this study has been made publicly available and easily accessible. Hence, researchers can easily back-calculate initial meropenem concentrations and reassess results from previous studies in which plasma samples were stored at −20 °C.

## 4. Materials and Methods

### 4.1. Sample Preparation

Plasma samples were prepared in increasing concentrations (0.5, 5, 20, 40, 60, 80, and 100 mg/L) by spiking drug-free plasma with an appropriate volume (0.05%, 0.5%, 2%, 4%, 6%, 8%, and 10% of the starting solution) of the meropenem working solution (1 mg/mL), that was freshly prepared by dissolving meropenem trihydrate in water for injection. For each concentration (i.e., E1–E7) and per time point (see Section 4.2), three separate plasma sample aliquots were prepared. Plasma samples were prepared at 2–8 °C and were immediately frozen after preparation (i.e., within 4 h). Meropenem trihydrate (certified reference material, purity ≥98%) was purchased from Sigma-Aldrich (Merck, Darmstadt, Germany). Drug-free plasma for experimental purposes was obtained from the Belgian Red Cross—Flanders (Dienst voor het bloed, Mechelen, Belgium).

### 4.2. Degradation Experiment Design

Stability of meropenem (base, corresponding to 87.6% of meropenem trihydrate) in human plasma at −20 °C was investigated at seven concentrations (0.44, 4.38, 17.5, 35.1, 52.6, 70.1, and 87.6 mg/L). This range of concentrations is representative for relevant concentrations (covering trough to peak concentrations) encountered in clinical practice with commonly used dosing regimens for meropenem (i.e., 1 g or 2 g q8 h as intermittent, extended or continuous infusion) [14,16]. For each concentration, samples (*n* = 3 per time point) were stored for 0, 7, 14, 21, 28, 42, 56, 70, 84, 112, 140, 168, 196, 224, 252, 280, 308, 336, and 364 days at −20 °C before being transferred on ice for a maximum of 5 min to −80 °C until analysis. Additionally, for each time point, blank (i.e., drug-free) plasma samples were added. None of the samples were subjected to a freeze–thaw cycle before quantification. All samples were prepared, and concentrations were measured in triplicate. All plasma samples were analyzed in batch at the end of the experiment by liquid chromatography coupled with tandem mass spectrometry. The detailed bioanalytical method is provided in the Appendix A.

For the stability experiment, measurements within 85–115% of the initial concentration were considered acceptable, as recommended in guidelines for bioanalytical method validation, with an exception of 80–120% for the lowest level of quantification [21].

### 4.3. Degradation Modeling

Degraded concentrations were modeled using regression analysis to approximate an underlying function f that maps the predictor variables, i.e., the storage period at −20 °C (days) and the measured concentration (mg/L), to the dependent variable, the relative meropenem concentration after degradation. All model simulations were performed using the mean (*n* = 3) measured meropenem concentrations. Relative meropenem concentration remaining after degradation was defined as the ratio between the measured concentration and the initial spiked concentration at t0.
$$meropenem\;degradation=\frac{C_{t}}{C_{0}},$$
where $C_{t}$ is the measured meropenem concentration (mg/L) at time t (days), and $C_{0}$ is the initial spiked concentration (mg/L). The predictor variables were standardized by subtracting the mean from each variable (centering) and then dividing the centered variable by its standard deviation (scaling), formalized as
$$X_{n}^{stand}=\frac{X_{i}-\overset{´}{X}}{\sigma },$$
where $\overset{´}{X}$ is the mean and σ is the standard deviation of the predictor variable. In a second-order quadratic model, centering reduces the correlation between the linear and quadratic terms [26], while scaling puts all predictors on a common scale [27]. For artificial neural networks, it is well established that standardizing the predictor variables to zero mean and unit variance usually results in faster convergence [28]. The two predictor variables were standardized independently and without exposure to the test distribution by computing the mean and standard deviation on the samples in the training dataset. The computed mean and standard deviation were then used to transform the predictor variables of the test dataset.

### 4.4. Polynomial Regression

Ordinary least-squares polynomial regression was used to model the degradation as a second-order, polynomial equation of the following form [29]:$$Y=b_{0}+\sum^{\text{​}}b_{i}X_{i}+\sum^{\text{​}}b_{ii}X_{i}^{2}+\sum^{\text{​}}b_{ij}X_{i}X_{j},$$
where Y is the predicted dependent variable, $X_{ij}$ represents the predictor variables, $b_{0}$ is the intercept, $b_{i}$ is the linear effect, $b_{ii}$ is the square effect, and $b_{ij}$ is the interaction effect. The mathematical relationship between the dependent variable Y and the two predictor variables could then be approximated by the quadratic (second-order) polynomial of the following form:$$Y=b_{0}+b_{1}X_{1}+b_{2}X_{2}+b_{11}X_{1}^{2}+b_{22}X_{2}^{2}+b_{12}X_{1}X_{2},$$

The fitting approximation was statistically evaluated with the F-statistic. The F-statistic assesses whether the polynomial (in this case quadratic) model explains the data better than an intercept-only model. Separate *t*-tests between each predictor variable and the dependent variable were performed to assess the significance (defined as *p* < 0.05) of each predictor term.

Simulations were carried out to investigate whether the degradation was concentration dependent. The degradation was considered significant when the lower 95% confidence interval exceeded the threshold of 85% of the initial spiked meropenem concentration.

### 4.5. Artificial Neural Networks

ANNs are a class of universal approximators that, in theory, can be used to approximate any function [30]. The network learns to approximate the function f by adjusting the interconnection weights between layers. Approximation of the weights is a non-convex optimization problem that aims to minimize the difference between the experimental measurements and the model predictions. To approximate the nonlinear degradation of meropenem, this study used the hyperbolic tangent (tanh) nonlinearity, formally defined as
$$tanh\left(x\right)=\frac{e^{x}-e^{-x}}{e^{x}-e^{-x}}.$$

The multilayer feed forward backpropagation network was optimized with the limited-memory Broyden–Fletcher–Goldfarb–Shanno algorithm [31], an optimization algorithm in the family of quasi-Newton methods. The proposed model consists of one to two hidden layers, each with 2–32 hidden neurons. The input layer contains two neurons, which provide each of the predictor variables to the second layer. The output layer contains one neuron, which corresponds to the predicted meropenem degradation. To improve generalization to unseen data, an L2 regularization penalty of 0.0001 to 0.1 was used. The ANN parameters were optimized by minimizing the mean square error (*MSE*), formally defined as
$$MSE=\frac{1}{n}\overset{n}{\underset{i=1}{\sum }}{\left(y_{i}-{\hat{y}}_{i}\right)}^{2},$$
where $y_{i}$ represents the observations, ${\hat{y}}_{i}$ represents the predicted values, and n is the number of observations available for analysis. The number of hidden layers, the number of hidden neurons, and the size of the L2 regularization term were the hyperparameters to be optimized.

### 4.6. Statistical Analysis

The ANN was validated by following a nested leave-one-experiment-out cross-validation approach, as visualized in Figure 5. This choice is warranted for two reasons: (1) by using a leave-one-experiment-out approach, the generalization of the model to all previously unseen concentrations within the studied range is assessed; (2) by nesting the cross validation folds into an inner and outer loop, there is separation among model selection, model fitting, and model evaluation. Without nesting the cross-validation folds, the model would be evaluated on the same data that were used to find the best hyperparameters. The bias resulting from fitting a model selection criterion with a non-nested cross-validation approach can be significantly large [32]. Since the second-order polynomial model has no model selection criterion, the performance was assessed using a non-nested leave-one-experiment-out cross-validation approach, consisting of only the outer loop.

The two regression models were quantitatively validated in terms of the *RMSE* and coefficient of determination (*R*^2^), which are formally defined as
$$RMSE=\sqrt{\frac{1}{n}\overset{n}{\underset{i=1}{\sum }}{\left(y_{i}-{\hat{y}}_{i}\right)}^{2}},\;R^{2}=1-\frac{\sum_{i=1}^{n}{\left(y_{i}-{\hat{y}}_{i}\right)}^{2}}{\sum_{i=1}^{n}{\left(y_{i}-{\bar{y}}_{i}\right)}^{2}},$$
where $y_{i}$ represents the observations, ${\bar{y}}_{i}$ represents the means of the observations, ${\hat{y}}_{i}$ represents the predicted values, and n is the number of observations available for analysis. To allow for better interpretability, the models were assessed in terms of percentage meropenem remaining (Y × 100).

The two regression models were implemented using Python version 3.7 and scikit-learn version 0.22.1 [33]. Statistical analysis was conducted using the statsmodels python library version 0.10.0 [34] and R statistical software version 3.5.3.

Further statistical analysis was performed to evaluate the goodness of fit for each model using the correlation between the observations and predicted values. Two-sided paired *t*-tests were computed to check the null hypothesis that the predictions and observations are not linearly related. However, the correlation describes the linear relationship between the model predictions and the observations but not their agreement [35]. Therefore, a Bland–Altman analysis was performed to assess the agreement by examining potential bias, the width of the limits of agreement, and variability proportional to the magnitude of observation (heteroscedasticity) [35]. Two-sided paired *t*-tests were computed to check the null hypothesis that the mean difference between the predictions and observations is zero. The significance level for all tests was set at 0.05.

## Figures and Tables

**Figure 1 antibiotics-10-00449-f001:**
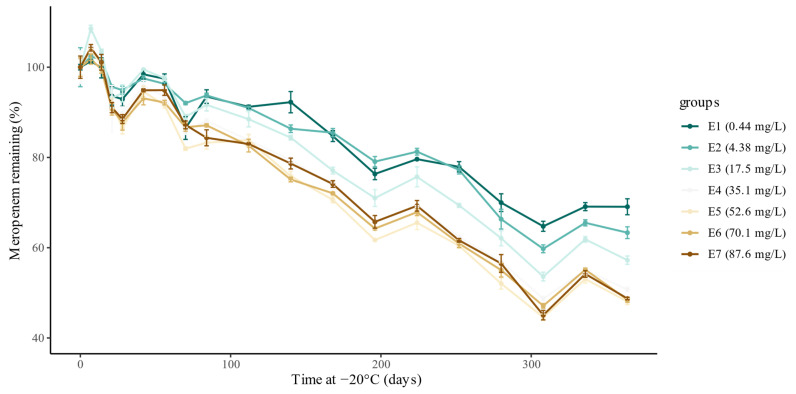
Experimentally measured degradation relative to the initial concentration (%), for each spiked concentration. The dots represent the mean concentration (%) in the three samples, and the bars represent the standard error for each mean.

**Figure 2 antibiotics-10-00449-f002:**
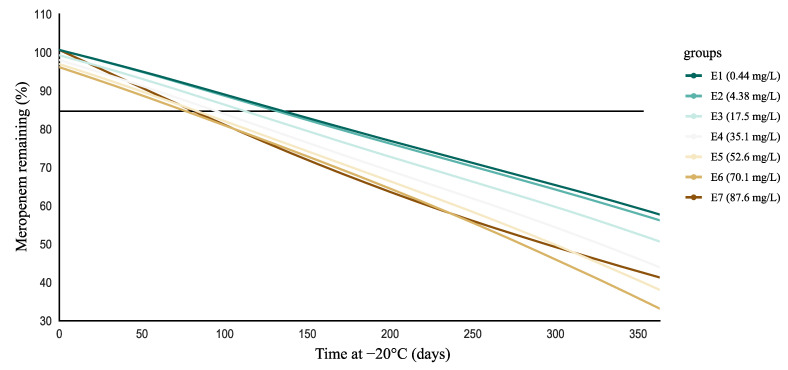
Simulation to assess the duration at which the lower 95% confidence interval crosses the critical threshold of 85% degradation (horizontal black line). Critical threshold values are shown in Appendix A (Appendix A). It can be observed that significant degradation occurred relatively more rapidly with increasing concentrations. This effect appeared to stabilize at higher concentrations, resulting in the degradation becoming significant at around 80 days for concentrations above 35.1 mg/L.

**Figure 3 antibiotics-10-00449-f003:**
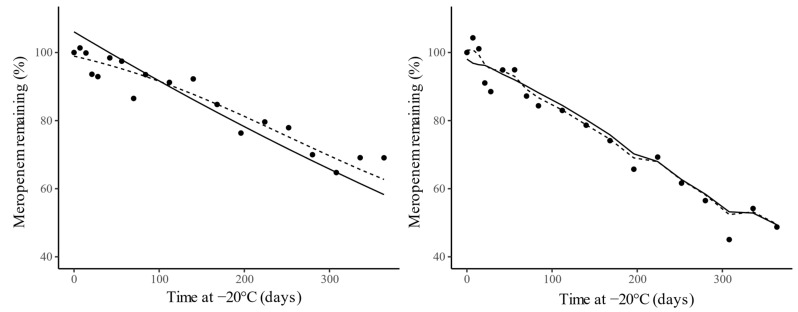
Comparison between the experimentally measured degradation (points) and the predicted values of the polynomial model (solid line) and neural network (dotted line) for experiment 1 (E1—0.44 mg/L; left) and experiment 7 (E7—87.6 mg/L; right), with spiked Meropenem concentrations of 0.44 mg/L and 87.6 mg/L, respectively.

**Figure 4 antibiotics-10-00449-f004:**
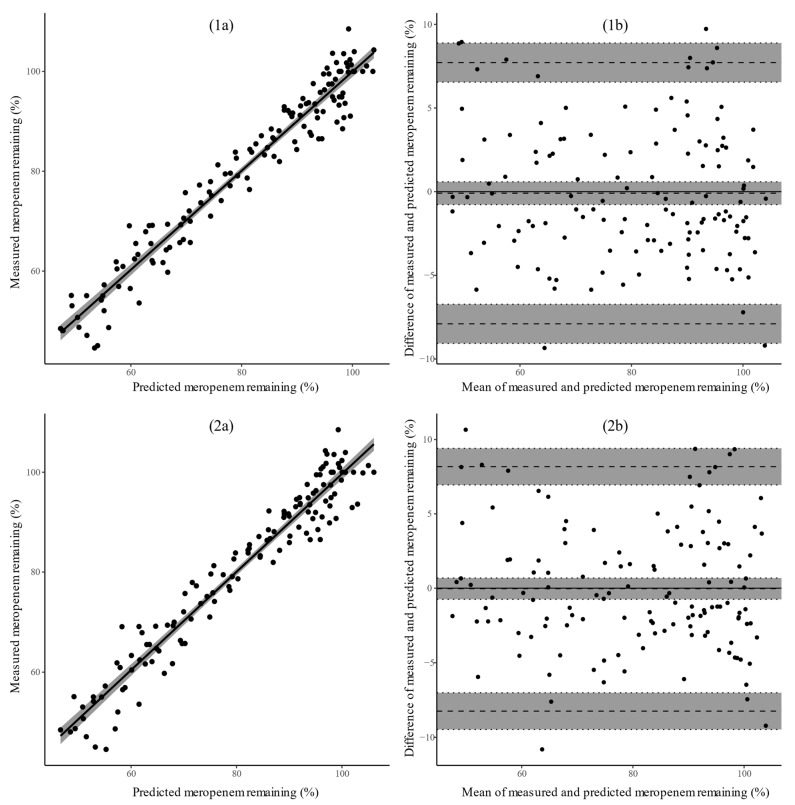
Assessment of the correlation (**left**), *r*, between the predicted degradation and experimentally measured degradation for the ANN (**1a**) and polynomial model (**2a**). The straight black line corresponds to the best linear fit, while the gray shaded area visualizes the 95% confidence interval. Two-sided *t*-tests were computed to evaluate the null hypothesis that the predictions and observations are not linearly related. Assessment of the agreement (**right**) between the predicted degradation and experimentally measured degradation for the ANN (**1b**) and polynomial model (**2b**) by means of a Bland–Altman plot. The dashed horizontal lines correspond to the bias and limits of agreement (±1.96 SD), while the gray shaded area visualizes the 95% confidence intervals. Two-sided *t*-tests were computed to evaluate the null hypothesis that the mean difference between the predictions and observations is zero (no bias). All results were derived from predictions on the left-out test experiment, i.e., measured concentrations that the model has never seen (see Section 4.6).

**Figure 5 antibiotics-10-00449-f005:**
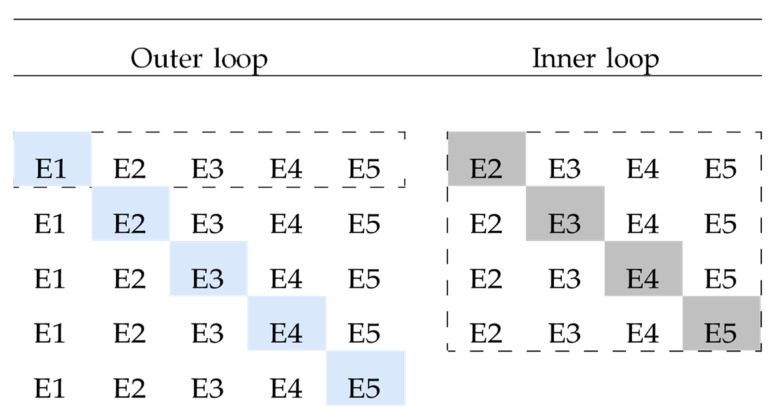
A visual overview of the nested leave-one-experiment-out cross-validation used to optimize and evaluate the artificial neural network. The nested procedure consisted out of an outer loop and inner loop. The hyperparameters were adjusted in the inner loop to optimize a model selection criterion. The weights were adjusted in the outer loop to optimize a model fitting criterion. The visualization is limited to five experiments (E1–E5). The dashed line denotes that the visualization is given for a single iteration of the outer loop, visualizing the tuning procedure for left-out test experiment E1. For this single iteration of the outer loop, experiment 1 (E1) was left out as a true holdout test set. The remaining experiments (E2–E5) were iteratively used as a holdout validation set to optimize the network hyperparameters in the inner loop. The hyperparameter set that resulted in the lowest mean squared error (MSE) was used to fit the model on all experiments of the outer loop (E2–E5). Lastly, this trained model was used to evaluate the model predictions on the left-out test experiment (E1). This conservative approach ensures separation among model selection, model fitting, and model evaluation.

**Table 1 antibiotics-10-00449-t001:** Mean meropenem plasma concentrations for spiked plasma samples stored at −20 °C for different durations over 1 year (*n* = 3 for each mean concentration reported).

Spiked Concentration (mg/L)	Time at −20 °C (Days)																		
	0	7	14	21	28	42	56	70	84	112	140	168	196	224	252	280	308	336	364
0.44 (E1)	0.46	0.46	0.46	0.43	0.43	0.45	0.45	0.40	0.43	0.42	0.42	0.39	0.35	0.37	0.36	0.32	0.30	0.32	0.32
4.38 (E2)	4.37	4.47	4.44	4.18	4.15	4.26	4.21	4.02	4.10	3.97	3.77	3.74	3.46	3.55	3.38	2.90	2.61	2.86	2.77
17.5 (E3)	16.4	17.8	17.0	15.3	15.5	16.3	16.0	14.6	15.0	14.5	13.8	12.6	11.6	12.4	11.4	10.2	8.8	10.1	9.4
35.1 (E4)	33.2	34.5	33.5	30.2	29.9	31.8	31.1	28.5	29.3	27.9	26.4	24.5	21.8	23.0	20.7	18.9	16.2	18.3	16.9
52.6 (E5)	50.3	52.1	50.6	46.2	43.5	47.5	46.1	41.2	41.9	42.1	38.1	35.5	31.0	32.9	30.4	26.2	22.4	26.7	24.2
70.1 (E6)	67.5	68.7	67.2	61.2	59.3	62.8	62.2	58.5	58.8	55.8	50.7	48.6	43.4	45.8	41.1	37.1	31.8	37.2	32.7
87.6 (E7)	83.3	86.9	84.2	75.8	73.7	79.0	79.0	72.6	70.2	69.1	65.5	61.7	54.7	57.7	51.3	47.1	37.5	45.1	40.6

**Table 2 antibiotics-10-00449-t002:** Results of the ordinary least-squares polynomial regression (POLY) and artificial neural network (ANN). All statistics and metrics were derived from predictions on the left-out test experiment, i.e., measured concentrations that the model has never seen. Means of the metrics are provided with their standard deviation (SD). The root-mean-square error (RMSE) between the predictions and experimental measurements was less than 6% for all experiments.

	POLY		ANN	
Experiment Number	RMSE (%)	*R* ^2^	RMSE (%)	*R* ^2^
E1 (0.44 mg/L)	5.68	0.868	3.60	0.909
E2 (4.38 mg/L)	3.03	0.948	2.91	0.949
E3 (17.5 mg/L)	4.24	0.922	4.15	0.926
E4 (35.1 mg/L)	3.89	0.950	3.75	0.953
E5 (52.6 mg/L)	4.49	0.928	4.25	0.941
E6 (70.1 mg/L)	3.43	0.962	3.14	0.967
E7 (87.6 mg/L)	3.93	0.945	3.03	0.970
Mean ± SD	4.10 ± 0.850	0.932 ± 0.031	3.55 ± 0.539	0.945 ± 0.022

**Table 3 antibiotics-10-00449-t003:** Polynomial regression coefficients. The coefficients are reported with the standard deviation (SD) and 95% confidence interval (CI). Two-sided *t*-tests were computed to evaluate the null hypothesis that the coefficient is equal to zero. The results suggest that the null hypothesis should be rejected (*p* < 0.05) for all predictors, except for the second-order effect of storage duration, which was found to be insignificant (*p* > 0.05).

	Coefficient (95% CI)	SD	*t*-Value	*p*-Value
$b_{0}$	0.779 (0.765–0.794)	0.007	106	0.000
$b_{1}$	−0.166 (−0.176–−0.156)	0.005	−34.5	0.000
$b_{2}$	−0.046 (−0.054–−0.037)	0.005	−10.0	0.000
$b_{11}$	0.004 (−0.006–0.013)	0.005	0.736	0.463
$b_{22}$	−0.021 (−0.032–−0.010)	0.005	−3.90	0.000
$b_{12}$	0.019 (0.010–0.028)	0.005	4.04	0.000

## Data Availability

The data represented in this study are available in the article or Appendix A.

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
