# Peer review of "Meropenem Stability in Human Plasma at −20 °C: Detailed Assessment of Degradation"

_antibiotics, 2021, doi:10.3390/antibiotics10040449_

Round 1

Reviewer 1 Report

Gijsen et al present an in vitro stability study of meropenem at -20°C. The broad-spectrum antibiotic meropenem is administered in many critically ill patients, and is the most frequently monitored of all beta-lactam antibiotics by means of TDM.  Accordingly, adequate stability is of uppermost importance. The authors performed a long-term stability test in the present work, given that not all laboratories have storage facilities at -80°C and several scientific groups failed to report stability data in their studies. 

The work is coherently written, and results are presented conclusively. However, certain aspects should be addressed before the manuscript can be accepted: 

Major aspects: 

  1. The degradation profile of E8 in figure 2 does not seem coherent, and does not agree with concentration-dependent degradation of E1-E6. A detailed explanation is missing. 
  2. The authors measured the samples as a batch series at the end of the experimental series - transferring samples from -20°C to -80°C. In figure 1, the experiment shows clear and uniform deviations for all concentration levels for given days (e.g. d7, d308). The authors do very briefly mention this fact in lines 277-278. It appears like a systematic error for these days (e.g. incorrect volumes filled for time-specific aliquots). Please explain the methods section in more detail to clarify ambiguities of the quality of your data, e.g. were samples prepared and processed in triplicate - or did you simply inject a samples 3 times into the LC-MS/MS system? Also standard deviations in the graph might be useful. 
  3. Commercially available QCs have become available for the TDM of beta-lactams, e.g. Chromsystems, Eureka. Have to authors included such QCs to confirm proper calibration and concentration ranges? Please clarify. 
  4. Stability testing should be done with samples that are as similar to native samples as possible. Did the authors also test long-term stability of frozen native samples. If not, this should be named as a further limitation. 
  5. Some authors that also used LC-MS/MS for stability testing, including Zander et al 2016 (doi: 10.1515/cclm-2015-0325) report very high degradation rates of meropenem, where they argue that only minor amounts of initial concentrations are left after 80-90 days at -20°C. Given that results may contradict each other, the discussion should be more extensive.

Minor aspects: 

  1. An reference addressing of the stability at -80°C should be added to the manuscript, given that stability must already be provided in the present experimental design. 
  2. Certain sections were not removed from the template tempalte, e.g. line 180-184, line 398. 
  3. Regarding the stability guidelines, lines 311-313. which guideline are the authors referring to. Please cite the EMA &/od FDA guideline/s of bioanalytic method validation.
  4. Accuracy and imprecision of your LC-MS/MS method should be added to the manuscript or supplementary. 
  5. For meropenem TDM the fastest possible TAT (typically within 8 hours) is required (e.g. for continous infusion regimens). Given that manufacturers stabilize their calibrators and QCs, the use of the results is mainly for study samples, but hardly for clinical samples that are processed as quickly as possible, compare lines 52-55. 
  6. Please report the purity of the Meropenem Trihydrate; include the information whether it is a certified reference material or not. 
  7. Supplementary: The supplementary states "Calibration curve standards were freshly spiked for every batch", but the way it is written in the main manuscript it appears that measurement was done as in a single batch after one year of storage. 
  8. Supplementary and main manuscript: Please describe how you prepared the working solution and how you prepared your stability sample; please clarify the percentage of working solution volume you used to prepare plasma samples (by spiking).  

Reviewer 2 Report

Gijsen M et al have conducted an interesting and well-done study on meropenem stability in human plasma at -20°C.

I would like to really congratulate the authors for performing such an interesting and well-done study on a needed topic. The paper is well-written, is easy to follow, the methods are clear and the results and conclussions are really important in clinical practice. The model is really interesting.

The English is adequate and the Tables and Figures are well-presented. 

Reviewer 3 Report

Comments to the Authors:

The paper focuses on “Meropenem stability in human plasma at -20 °C”. The present manuscript is a compilation of Human plasma stability data for the compound Meropenem at -20 degree Celsius for over one year. The manuscript is statistically sound.

I have a few questions to be answered before the manuscript can be accepted for publication. Please find my comments below:

  1. The fundamental question that may arise in the reader's mind mere reading the manuscript title & abstract is why there is a need to conduct a stability study in only -20 degree C while the study can be simultaneously carried out at -80 degree C as well? The stability data in -80 degree C and -20 degree C may give a clear picture and provide comparative data to the researchers to choose among the best two. Moreover, one reference the authors have quoted in line 62 that if the samples to be stored for more than three days has to be kept at -80 degree C for analysis, then there is a strong need to choose -80 degree C as one reference point for plasma stability study for Meropenem.
  2. In most cases, the compounds are more stable in the biological matrix when kept at -80 degree C compared to -20 degree C. If that is the case with this compound too, then what is the need for storing the human clinical samples at -20 degree C when the -80 Deepfreeze are universally available in all the bioanalytical labs and a preferred choice for storage?
  3. Although the manuscript is statistically sound however the data provided for the study is not comprehensive. The degradation kinetics data is missing, and there is no discussion of plasma half-life anywhere in the manuscript.
  4. I just wish to confirm if the authors have maintained the one-time freeze-thaw cycle for the samples analyzed for the plasma stability study.
  5. There is no mention of positive and negative controls for the stability assay. Please provide details in the manuscript.
  6. Is the bioanalytical method used to analyze Meropenem samples was optimized for the first time or a previously reported method? If this method is previously reported, please provide a suitable reference; else, give more details about the LCMS method, including validation data and quality controls used to validate the assay.
  7. Minor correction: In the supplementary material in the bioanalytical method section, the authors have used “,” instead of “.”for Meropenem transitions. Please correct.
  8. What are the concentrations of the QC samples? Please provide QC sample analysis data in the manuscript/supplementary.
  9. Several key bioanalytical method information is missing. The author should provide the make and model of LCMS, details of protein precipitation method, MS parameters, and column info.
  10. The author should include more information in the conclusion and the discussion section regarding the correlation of the stability data at -20 degree C with the AUC and the clearance data from the literature.
  11. The authors should have conducted a similar study in phosphate buffer saline to conclude whether the degradation is from the plasma components or not.
  12. How have you prepared the standard stock of Meropenem? If the standard stock solution was prepared in DMSO, what was the spiking DMSO percent in human plasma? Please provide details in the manuscript as DMSO plays a critical role in stability assay.

Reviewer 4 Report

General evaluation

The research carried out in this manuscript is relevant and could be of potential interest for the Antibiotics journal readers. The information is properly presented and the results are adequately shown. The authors present a detailed description-model of meropenem degradation at -20°C, based on a large number of samples over one year and a clinical range of meropenem concentrations. In addition, the model provided in the manuscript is publicly available and allows other researchers to re-asses meropenem concentration which samples were stored at -20°C. In my opinion, the manuscript is valuable and relevant. However, additional few clarifications would be highly appreciated to better understand specific details of the results presented in this manuscript.

Specific comments:

INTRODUCTION

I1. Line 54: Please include reference of B-lactam studies referred to.

METHODS:

M1. Lines 297-313: As a wide range of meropenem concentrations were analysed, carry-over phenomenon of the HPLC analytical method should be studied. Did the authors validate the analytical method for the carry-over effect (FDA and/or EMA recommendations)? In addition, were meropenem samples analysed in a specific order (i.e. form small to high concentrations)?

M2. Lines 297-313: Freeze/thaw cycles can influence on drug degradation and stability, therefore, freeze/thaw stability must be evaluated when the same sample is used for several analyses. Meropenem concentrations evaluated (n=7) were freeze and thaw at each analysis time (0, 7, 21, etc. days) or they were storage as aliquots? This details must be included in the text.

M3. Lines 301: meropenem doses taken into account to project the representative range of concentrations must be included.

M4. Lines 342: Did the authors evaluate reduced models such as liner regression (with/without ponderation) or power relationship models instead the polynomial regression?

RESULTS

R1. Figure 4. Inclusion of 1a, 1b, 2a, 2b within the figure would be recommended for better understanding of the figure caption.

DISCUSSION

D1. Lines 181-184. First paragraph of discussion section looks like a guideline text of this section. In my opinion, this paragraph should be removed.

D2. Lines 201-208. In the specific case of the lower concentrations, close to the LLOQ of the analytical method, a 20% precision and accuracy is accepted for validation purposes of analytical methods in biological samples. Thus, the time to reach a degradation threshold of 80% recovery of the original concentration (in addition to the current 15% thershold) can be taken into account as this variability could also be imputed to the precision/accurate of the analytical method at these low concentrations. This would increase the validity of meropenem samples at very low concentrations up to aproximately 160-180 days since storage at -20°C (figure 2). I would recommend to discuss this fact in addition of the current valid arguments in the manuscript.

Round 2

Reviewer 1 Report

All relevant points have been revised. 

For the purpose of transparency, i highly recommend that the authors indicate contract research as such at the time of (future) initial submission, and not at later stages of a given review process. 

Reviewer 3 Report

I appreciate the authors for providing satisfactory answers to all the 12 points. The manuscript looks more comprehensive now, with detailed information in the supplementary supporting the Manuscript.

Still, I am partially satisfied with point 5:

Point 5: There is no mention of positive and negative controls for the stability assay. Please provide details in the manuscript.
Response 5: In the design of the degradation experiment, we included blanks as negative controls. These were samples spiked with drug-free plasma. As positive control, we included plasma samples that were immediately stored at -80°C (i.e., 0 days at -20°C).

I am happy with the above response from the authors but let me reframe my question. Have you used positive or negative control (Quality controls) to validate the stability assay? It is critical to provide QC controls (either positive or negative control) with a set of test compounds in each assay, with a known plasma t1/2. This ensures that the assay is working properly and that the plasma activity meets the defined requirements. A few QC compounds have been proposed, e.g., Diltiazem, Vinpocetine, Poldine, Benfluorex [Di, E.H. Kerns, Y. Hong, H. Chen, Development and application of high throughput plasma stability assay for drug discovery, Int. J. Pharm. 297 (2005) 110–119].
